

# Light-harvesting chlorophyll a/b-binding protein-coding genes in jatropha and the comparison with castor, cassava and arabidopsis

Yongguo Zhao[1,2], Hua Kong[2], Yunling Guo[2] and Zhi Zou[2]

[1] Guangdong University of Petrochemical Technology, Maoming, China
[2] Key Laboratory of Biology and Genetic Resources of Tropical Crops, Ministry of Agriculture and Rural Affairs, Hainan Key Laboratory for Biosafety Monitoring and Molecular Breeding in Off-Season Reproduction Regions, Institute of Tropical Biosciences and Biotechnology, Chinese Academy of Tropical Agricultural Science, Haikou, China

## ABSTRACT

The Lhc (light-harvesting chlorophyll a/b-binding protein) superfamily represents a class of antennae proteins that play indispensable roles in capture of solar energy as well as photoprotection under stress conditions. Despite their importance, little information has been available beyond model plants. In this study, we presents a first genome-wide analysis of *Lhc* superfamily genes in jatropha (*Jatropha curcas* L., Euphorbiaceae), an oil-bearing plant for biodiesel purpose. A total of 27 members were identified from the jatropha genome, which were shown to distribute over nine out of the 11 chromosomes. The superfamily number is comparable to 28 present in castor (*Ricinus communis*, Euphorbiaceae), but relatively less than 35 in cassava (*Manihot esculenta*, Euphorbiaceae) and 34 in arabidopsis (*Arabidopsis thaliana*) that experienced one or two recent whole-genome duplications (WGDs), respectively. In contrast to a high number of paralogs present in cassava and arabidopsis, few duplicates were found in jatropha as observed in castor, corresponding to no recent WGD occurred in these two species. Nevertheless, 26 orthologous groups representing four defined families were found in jatropha, and nearly one-to-one orthologous relationship was observed between jatropha and castor. By contrast, a novel group named SEP6 was shown to have been lost in arabidopsis. Global transcriptome profiling revealed a predominant expression pattern of most *JcLhc* superfamily genes in green tissues, reflecting their key roles in photosynthesis. Moreover, their expression profiles upon hormones, drought, and salt stresses were also investigated. These findings not only improve our knowledge on species-specific evolution of the *Lhc* supergene family, but also provide valuable information for further studies in jatropha.

Corresponding author
Zhi Zou, zouzhi@itbb.org.cn,
zouzhi2008@126.com

## INTRODUCTION

Light-harvesting chlorophyll a/b-binding protein (Lhc) superfamily, defined by the presence of a conserved chlorophyll-binding (CB) domain in the transmembrane alpha-helix, is composed of four distinct nuclear-encoded antennae protein families in green plants, i.e., Lhc, Lil (light-harvesting-like), PsbS (photosystem II subunit S), and FCII (ferrochelatase II) (*Klimmek et al., 2006*; *Zou & Yang, 2019a*). In contrast to an orphan group present in both PsbS and FCII families, the Lhc family, initially known as CAB (chlorophyll a/b-binding protein), contains two evolutionary groups named Lhca and Lhcb that are associated with photosystem I or II (PSI/II), respectively (*Jansson, 1999*; *Klimmek et al., 2006*). The Lil family includes four diverse subfamilies, i.e., OHP (one-helix protein), SEP (stress-enhanced protein), Lil1 or ELIP (early light-induced protein), and Lil8 or Psb33 (photosystem II protein 33) (*Engelken, Brinkmann & Adamska, 2010*; *Zou, 2018*). Among them, OHP and SEP can be further divided into several groups: the OHP subfamily includes two groups named OHP1/Lil2 and OHP2/Lil6, whereas the SEP subfamily contains six groups, i.e., SEP1/Lil4, SEP2/Lil5, SEP3/Lil3, SEP4, SEP5, and SEP6 (*Engelken, Brinkmann & Adamska, 2010*; *Zou & Yang, 2019a*). Investigation of their origin suggested that OHP is more primitive, which is more likely to result from the plastid-encoded HLIP (high light-induced protein) via gene transfer after the primary endosymbiosis (*Koziol et al., 2007*; *Engelken, Brinkmann & Adamska, 2010*). In addition to light harvesting and transport, growing evidence shows that *Lhc* superfamily members are also involved in regulation and distribution of excitation energy between PSI and PSII, maintenance of thylakoid membrane structure, photoprotection as well as response to various stresses (*Pan et al., 2011*; *Fan et al., 2015*; *Fristedt et al., 2015*; *Hey et al., 2017*; *Myouga et al., 2018*).

*Jatropha curcas* L. ($2n = 22$), commonly known as jatropha, physic nut, barbados nut, or purging nut, is a perennial large shrub or small tree (*Mazumdar et al., 2018*; *Zou et al., 2016*; *Zou et al., 2018*). Jatropha belongs to the Euphorbiaceae family, which also includes castor (also known as castor bean, *Ricinus communis* L.) and cassava (*Manihot esculenta* Crantz) and is characterized with high photosynthesis and high biomass (*Zou et al., 2018*; *Zou & Yang, 2019a*; *Zou & Yang, 2019b*). As a potential non-edible energy crop, jatropha produces high level of fossil fuel-like oil in its seeds, which can be easily processed into biodiesel (*Fairless, 2007*; *Berchmans & Hirata, 2008*; *Kumar & Sharma, 2008*; *Maghuly & Laimer, 2013*). Additionally, this species also has several unique characteristics like easy propagation, rapid growth, and adaptation to semiarid and barren soil environments (*Montes & Melchinger, 2016*). Although originated from Mesoamerica, jatropha can now be widely found in many tropical and subtropical countries of Africa and Asia (*Wu et al., 2015*; *Li et al., 2017*). Nevertheless, its commercial cultivation has failed mainly due to low productivity (*Montes & Melchinger, 2016*; *Mazumdar et al., 2018*). Thereby, uncovering the molecular mechanism underlying and characterization of genes involved in yield formation are prerequisites. In this study, we would like to present a first genome-wide analysis of the *Lhc* supergene family in jatropha, including gene structures, chromosome (Chr) locations, evolutionary relationships, sequence characteristics, global expression profiles as well as comprehensive comparison with arabidopsis, cassava, and castor. These results will not

only improve our knowledge on species-specific evolution of the *Lhc* supergene family, but also provide valuable information for further functional analysis in jatropha.

## MATERIALS & METHODS

### Identification of *Lhc* superfamily genes

As shown in Table S1, 34 *AtLhc* superfamily genes were retrieved from TAIR (https://www.arabidopsis.org/, Araport11) according to previous literatures. To facilitate evolutionary analysis, 28 and 35 superfamily members present in castor and cassava (see Table S1), two Euphorbiaceous plants, were also obtained from Phytozome (Version 12, https://phytozome.jgi.doe.gov/pz/portal.html). Homologs present in the jatropha genome (*Wu et al., 2015*) were identified via the tBLASTn (*Altschul et al., 1997*; *E*-value, 1e–5) search by using above protein sequences as queries. Gene models of candidates were revised via aligning mRNA to loci-encoding scaffolds. Presence of the conserved CB domain (PF00504) was checked using MOTIF Search (https://www.genome.jp/tools/motif/), and exon-intron structures were displayed using Gene Structure Display Server (GSDS 2.0, https://gsds.cbi.pku.edu.cn/). Putative transmembrane helix (TMH) was predicted using CCTOP (http://cctop.enzim.ttk.mta.hu/) as well as sequence alignment. Chloroplast transit peptide (TP) of deduced proteins and biochemical parameters of mature peptides were determined using ChloroP (Version 1.1, https://www.cbs.dtu.dk/services/ChloroP/) and ProtParam (https://web.expasy.org/protparam/), respectively.

### Chromosome location and synteny analysis

Gene distribution on chromosomes was analyzed using MAPchart 2.3 (*Voorrips, 2002*). For synteny analysis, the all-to-all BLASTP method was used to identify duplicate pairs, and MicroSyn (*Cai et al., 2011*) was used to detect microsynteny. Orthologs across different species were inferred from the best-reciprocal-hit (BRH)-based BLAST analysis as well as synteny analysis for jatropha and castor.

### Sequence alignment, phylogenetic, and conserved motif analyses

Multiple sequence alignment was carried out using MUSCLE (*Edgar, 2004*). Phylogenetic tree construction was performed using MEGA7 (*Kumar, Stecher & Tamura, 2016*) with the maximum likelihood method (bootstrap: 1,000). Conserved motifs were identified using MEME (https://meme-suite.org/tools/meme): any number of repetitions; maximum number of motifs, 25; minimum sites, 2; and, the optimum width of each motif, between 6 and 100 residues.

### Gene expression analysis

Transcriptome datasets used for expression profiling are shown in Table S2. Except for tissue-specific transcriptomes, other samples were performed for at least two biological replicates. Quality control of raw reads was carried out using fastQC (https://www. bioinformatics.babraham.ac.uk/projects/fastqc/). Read mapping were performed using Bowtie 2 (*Langmead & Salzberg, 2012*), and the relative transcript level of each gene was presented as FPKM (fragments per kilobase of exon per million fragments

mapped, for pair-ended samples) or RPKM (Reads per kilobase per million mapped reads, for single-ended samples) (*Mortazavi et al., 2008*). RSEM (v1.2.27) (*Li & Dewey, 2011*) with parameters "log2Ratio ≥ 1" and "FDR <0.001" were used to determine differentially expressed genes.

## RESULTS

### Identification and chromosome locations of 27 *Lhc* superfamily genes in jatropha

The BLAST search resulted in 27 *JcLhc* superfamily genes from the jatropha genome (*Wu et al., 2015*), which represent four previously defined families (i.e., *Lhc*, *Lil*, *PsbS*, and *FCII*) or eight subfamilies (i.e., *Lhca*, *Lhcb*, *PsbS*, *OHP*, *SEP*, *ELIP*, *Psb33*, and *FCII*). Each subfamily contains one to nine members that were named after their orthologs in castor (see below), i.e., *JcLhca1–6*, *JcLhcb1.1–1.2*, *JcLhcb2–8*, *JcPsbS*, *JcELIP*, *JcOHP1–2*, *JcSEP1–6*, *JcPsb33*, and *JcFCII*, respectively. These genes were shown to distribute over 25 scaffolds. Although most scaffolds contain a single member, two of them harbor two, i.e., scaffold160 (i.e., *JcLhcb1.1* and *JcLhcb1.2*) and scaffold211 (i.e., *JcLhca6* and *JcSEP5*) (Table 1). With the help of 1,208 available genetic markers, these genes were further anchored to nine chromosomes, and the gene number of each chromosome varies from one to five (Fig. 1).

The expression of all *JcLhc* genes was supported by available Sanger sequencing-derived expressed sequence tags (ESTs) and/or RNA sequencing (RNA-seq), where *JcLhcb1.1* harbors the maximum of 28 EST hits. The intron number of these genes varies from zero to nine: approximately 11.11% of genes are intronless, and 29.63%, 22.22%, 11.11%, 11.11%, 11.11% or 3.70% contain two, one, three, four, five and nine introns, respectively (Table 1 and Fig. 2). Similar exon-intron structure was also observed in castor, cassava, and arabidopsis (see Table S1), implying a conserved evolution between these species. The average length of coding sequences (CDS) is about 760 bp, varying from 357 bp of *JcOHP1* to 1,500 bp of *JcFCII*. Compared with CDS, the intron length is relatively more variable, ranging from 79 bp of *JcLhcb6* to 8,100 bp of *JcFCII* and with the average length of 1,259 bp (Table 1 and Fig. 2). The CDS of *JcLhcb1.1* and *JcLhcb1.2*, which are reversely clustered on scaffold160, was shown to exhibit 96.7% identity. Thereby, they are more likely to result from tandem duplication (*Zou, Yang & Zhang, 2019*; *Zou & Yang, 2019a*; *Zou & Yang, 2019c*).

#### Synteny analysis and determination of orthologous groups

Orthologs of *JcLhc* superfamily genes in castor, cassava, and arabidopsis were further identified by using the BRH method, resulting in 26 orthologous groups (OGs) when the definition was confined to at least one member present in more than two species examined (Table 1). The result is highly consistent with phylogenetic analysis (see below) as well as synteny analysis performed between jatropha and castor, where one-to-one orthologous relationship was observed with exception of the Lhcb1 group with two-to-three (Table 1). Interestingly, *RcLhcb1.2*, which may originate by dispersed duplication, is located on scaffold30005 together with *RcLhca3*. However, no collinear gene was found for *RcLhcb1.2* in jatropha (see Fig. S1). By contrast, orthologous relationships between jatropha and

Peer J

**Table 1** 27 *Lhc* superfamily genes identified in jatropha.

| Subfamily | Gene name | Locus ID | Scaffold position | Nucleotide length (bp, from start to stop codons) | | Intron no. | EST no. | AA | TP length | TMH | Ortholog | | | OG |
|---|---|---|---|---|---|---|---|---|---|---|---|---|---|---|
| | | | | CDS | Gene | | | | | | Rc | Me | At | |
| | *JcLhca1* | JCGZ_23938 | scaffold794:74368-75803(−) | 738 | 1,076 | 3 | 0 | 245 | 45 | 3 | RcLhca1 | MeLhca1.1 MeLhca1.2 | AtLhca1 | Lhca1 |
| | *JcLhca2* | JCGZ_17961 | scaffold502:3176588-3178724(+) | 813 | 1,491 | 4 | 1 | 270 | 58 | 3 | RcLhca2 | MeLhca2.1 MeLhca2.2 | AtLhca2 | Lhca2 |
| | *JcLhca3* | JCGZ_15032 | scaffold42:178242-179765(−) | 816 | 1,171 | 2 | 8 | 271 | 38 | 3 | RcLhca3 | MeLhca3 | AtLhca3 | Lhca3 |
| | *JcLhca4* | JCGZ_11643 | scaffold328:2197266-2195814(−) | 750 | 917 | 2 | 2 | 249 | 48 | 3 | RcLhca4 | MeLhca4.1 MeLhca4.2 | AtLhca4 | Lhca4 |
| Lhca | *JcLhca5* | JCGZ_04265 | scaffold159:84206-85704(+) | 795 | 1,255 | 5 | 0 | 264 | 57 | 3 | RcLhca5 | MeLhca5 | AtLhca5 | Lhca5 |
| | *JcLhca6* | JCGZ_07509 | scaffold211:3089530-3091440(−) | 792 | 1,612 | 4 | 0 | 263 | 42 | 3 | RcLhca6 | MeLhca6 | AtLhca6 | Lhca6 |
| | *JcLhcb1.1* | JCGZ_04588 | scaffold160:101030-101541(+) | 798 | 798 | 0 | 28 | 265 | 35 | 3 | RcLhcb1.1 RcLhcb1.2 RcLhcb1.3 | MeLhcb1.1 MeLhcb1.2 MeLhcb1.3 | AtLhcb1.1 AtLhcb1.2 AtLhcb1.3 AtLhcb1.4 AtLhcb1.5 | Lhcb1 |
| | *JcLhcb1.2* | JCGZ_04587 | scaffold160:97668-99902(−) | 798 | 798 | 0 | 17 | 265 | 35 | 3 | RcLhcb1.1 RcLhcb1.2 RcLhcb1.3 | MeLhcb1.1 MeLhcb1.2 MeLhcb1.3 | AtLhcb1.1 AtLhcb1.2 AtLhcb1.3 AtLhcb1.4 AtLhcb1.5 | Lhcb1 |
| | *JcLhcb2* | JCGZ_18481 | scaffold529:239615-242657(−) | 798 | 2,225 | 1 | 10 | 265 | 37 | 3 | RcLhcb2 | MeLhcb2.1 MeLhcb2.2 | AtLhcb2.1 AtLhcb2.2 AtLhcb2.3 | Lhcb2 |
| | *JcLhcb3* | JCGZ_00703 | scaffold108:360030-361580(−) | 804 | 1,202 | 2 | 5 | 267 | 44 | 3 | RcLhcb3 | MeLhcb3 | AtLhcb3 | Lhcb3 |
| | *JcLhcb4* | JCGZ_25025 | scaffold843:295347-296708(+) | 858 | 957 | 1 | 8 | 285 | 31 | 3 | RcLhcb4 | MeLhcb4 | AtLhcb4.1 AtLhcb4.2 | Lhcb4 |
| Lhcb | *JcLhcb8* | JCGZ_01281 | scaffold11:738853-741122(+) | 840 | 1,786 | 1 | 0 | 279 | 31 | 3 | RcLhcb8 | MeLhcb8 | AtLhcb8 | Lhcb8 |
| | *JcLhcb5* | JCGZ_06701 | scaffold200:160436-162429(−) | 876 | 1,573 | 5 | 1 | 291 | 41 | 3 | RcLhcb5 | MeLhcb5 | AtLhcb5 | Lhcb5 |
| | *JcLhcb6* | JCGZ_20203 | scaffold645:375633-376811(+) | 765 | 844 | 1 | 1 | 254 | 49 | 3 | RcLhcb6 | MeLhcb6 | AtLhcb6 | Lhcb6 |
| | *JcLhcb7* | JCGZ_08016 | scaffold221:188978-192485(−) | 1,014 | 3,050 | 5 | 0 | 337 | 52 | 3 | RcLhcb7 | MeLhcb7 | AtLhcb7 | Lhcb7 |
| PsbS | *JcPsbS* | JCGZ_12094 | scaffold339:857142-859344(−) | 825 | 1,726 | 3 | 2 | 274 | 62 | 4 | RcPsbS | MePsbS | AtPsbS | PsbS |
| ELIP | *JcELIP* | JCGZ_02231 | scaffold119:1376949-1378080(−) | 585 | 808 | 2 | 6 | 194 | 86 | 3 | RcELIP | MeELIP | AtELIP1 AtELIP2 | ELIP |
| OHP | *JcOHP1* | JCGZ_09105 | scaffold250:2426033-2426931(−) | 357 | 552 | 2 | 2 | 118 | 48 | 1 | RcOHP1 | MeOHP1 | AtOHP1 | OHP1 |
| | *JcOHP2* | JCGZ_08332 | scaffold224:432321-435041(+) | 552 | 2,339 | 1 | 0 | 183 | 47 | 1 | RcOHP2 | MeOHP2.1 MeOHP2.2 | AtOHP2 | OHP2 |

*(continued on next page)*

Zhao et al. (2020), *PeerJ*, DOI 10.7717/peerj.8465

**Table 1** (*continued*)

| Subfamily | Gene name | Locus ID | Scaffold position | Nucleotide length (bp, from start to stop codons) | | Intron no. | EST no. | AA | TP length | TMH | Ortholog | | | OG |
|---|---|---|---|---|---|---|---|---|---|---|---|---|---|---|
| | | | | CDS | Gene | | | | | | Rc | Me | At | |
| | *JcSEP1* | JCGZ_23193 | scaffold7:432788-436707(−) | 438 | 3,498 | 3 | 3 | 145 | 71 | 2 | RcSEP1 | MeSEP1 | AtSEP1 | SEP1 |
| | *JcSEP2* | JCGZ_09398 | scaffold255:592063-593949(+) | 582 | 1,322 | 1 | 0 | 193 | 46 | 2 | RcSEP2 | MeSEP2 | AtSEP2 | SEP2 |
| | *JcSEP3* | JCGZ_03488 | scaffold137:779011-781250(+) | 780 | 1,919 | 2 | 4 | 259 | 100 | 2 | RcSEP3 | MeSEP3.1 MeSEP3.2 | AtSEP3.1 AtSEP3.2 | SEP3 |
| SEP | *JcSEP6* | JCGZ_26324 | scaffold906:2486216-2487759(−) | 759 | 932 | 2 | 0 | 252 | 88 | 2 | RcSEP6 | MeSEP6 | – | SEP6 |
| | *JcSEP4* | JCGZ_06634 | scaffold20:991705-992601(+) | 570 | 570 | 0 | 0 | 189 | 56 | 2 | RcSEP4 | MeSEP4 | AtSEP4 | SEP4 |
| | *JcSEP5* | JCGZ_07816 | scaffold211:5206555-5210867(−) | 447 | 3,909 | 4 | 0 | 148 | 75 | 1 | RcSEP5 | MeSEP5 | AtSEP5 | SEP5 |
| Psb33 | *JcPsb33* | JCGZ_17235 | scaffold5:830004-833169(+) | 867 | 2,803 | 2 | 1 | 288 | 62 | 1 | RcPsb33 | MePsb33.1 MePsb33.2 | AtPsb33 | Psb33 |
| FCII | *JcFCII* | JCGZ_24260 | scaffold813:61000-71385(−) | 1,500 | 9,600 | 9 | 0 | 499 | 86 | 1 | RcFCII | MeFCII | AtFCII | FCII |

**Notes.**

AA, amino acid; At, *Arabidopsis thaliana*; bp, base pair; CDS, coding sequence; EST, expressed sequence tag; Me, *Manihot esculenta*; OG, orthologous group; Rc, *Ricinus communis*; TMH, trans-membrane helix; TP, transit peptide).
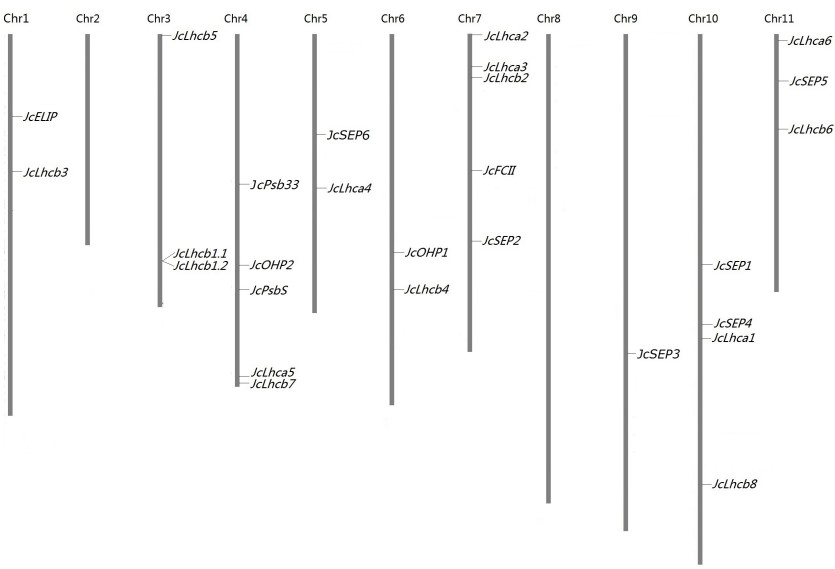

**Figure 1 Chromosomal locations of *JcLhc* superfamily genes.** Chromosome serial numbers are indicated at the top of each chromosome. Chr: chromosome.

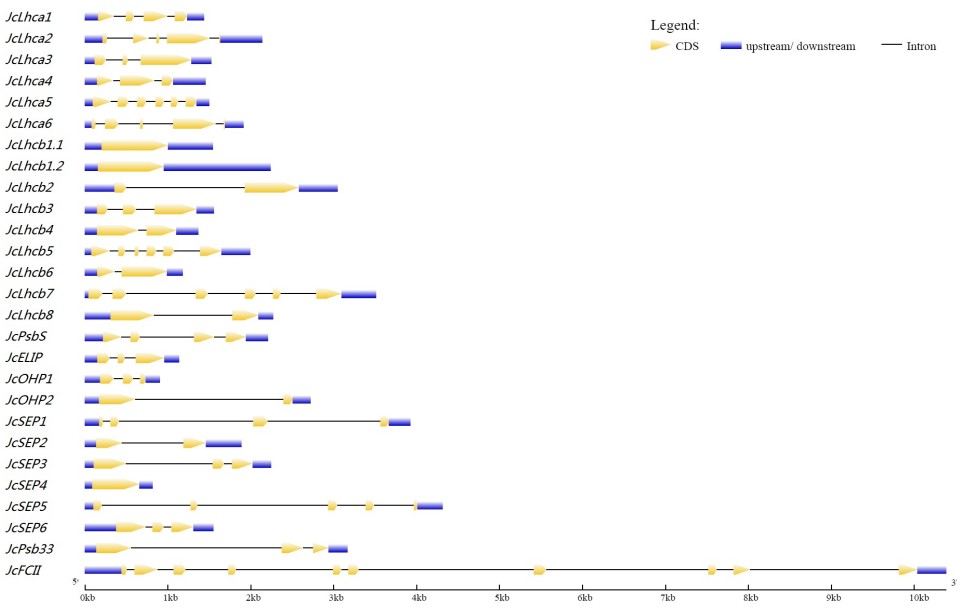

**Figure 2 Exon-intron structures of *JcLhc* superfamily genes.** The graphic representation of the gene models is displayed using GSDS. GSDS: gene structure display server.

cassava/arabidopsis are relatively complex, which include one-to-one, one-to-two, one-to-three, and two-to-five, corresponding to one or more recent WGDs occurred in these two species. It is worth noting that, SEP6, a recently identified group that is present in jatropha, castor, and cassava, is absent from arabidopsis (Table 1), implying species/lineage-specific

gene loss. Additionally, species-specific gene expansion was also observed: duplicates identified in jatropha (one) and castor (two) were shown to result from tandem or dispersed duplication, respectively; nine duplicates identified in cassava were derived from tandem duplication (three) and whole-genome duplication (WGD) (six); in arabidopsis, four or five duplicates were derived from tandem duplication and WGD, respectively (Table 1 and Table S1).

Although not exactly the same within a family, the exon-intron structure is highly conserved within a certain OG: Lhcb1 and SEP4 are intronless; one-intron-containing groups include Lhcb2/-4/-6/-8, OHP2, and SEP2, whereas two-intron groups include Lhca3/-4, Lhcb3, ELIP, OHP1, SEP3/-6, and Psb33; Lhca1 and SEP1 feature three introns, whereas Lhca2/-6, and SEP5 feature four introns; three groups (i.e., Lhcb5/-7 and Lhca5) contain five introns, whereas only one group (i.e., FCII) harbors nine introns (Fig. 2, Table 1, and Table S1).

## Phylogenetic analysis, sequence features, and conserved motifs

As shown in Table 1, the deduced JcLhc superfamily proteins consist of 118–499 amino acids (AA) with one to four TMHs (Fig. S2), and the predicted length of transit peptide ranges from 31 to 100 residues (Table 1). Several physical and chemical parameters of mature peptides were further calculated: the molecular weight (MW) and isoelectric point ($p$I) values of mature proteins in jatropha range from 7.55 to 46.76 kilodalton (kDa) or from 4.56 to 9.99, respectively; about 81.48% of JcLhc superfamily proteins harbor a $p$I value of less than 7, which is relatively less than 85.71% in castor, 91.43% in cassava, or 91.18% in arabidopsis; and, about 59.26% of JcLhc superfamily proteins harbor a grand average of hydropathicity (GRAVY) value of less than 0, which is relatively less than 64.29% in castor, 62.86% in cassava, or 73.53% in arabidopsis (Table 1 and Table S1).

Except for JcPsb33 that contains a CB-like domain (see Fig. S3), all other JcLhc superfamily proteins include the core CB domain of approximately 20 AA (see Table S1). Nevertheless, their overall sequence similarity was shown to be considerably low, even within the conserved Lhc family (ranging from 27.2% to 98.5%, see Table S3). To keep the analysis reliable, an independent phylogenetic tree was constructed for each subfamily by using full-length proteins from jatropha, castor, cassava, and arabidopsis. As shown in Fig. 3A, subfamilies Lhca, Lhcb, OHP, and SEP are clearly clustered into six, eight, two or six phylogenetic groups respectively, corresponding to 22 OGs as described above, i.e., Lhca1–6, Lhcb1–8, OHP1, OHP2, and SEP1–6. Among them, Lhcb8 and SEP6 exhibit a closer relationship with Lhcb4 and SEP3, with a similarity of 79.5% or 55.5%, respectively (Table S3), where JcLhcb8 harbors a relative shorter C-terminal in relation to JcLhcb4 (Fig. S2).

To reveal possible divergence of members within a certain OG and between different OGs/(sub)families, conserved motifs were analyzed using MEME. As shown in Fig. 3B and Table 2, motifs are highly variable between subfamilies or even between different evolutionary groups, and considerably more motifs were identified in the Lhc family as compared with PsbS, FCII, and Lil families. Among 25 motifs identified, Motif 1, which is characterized as the CB domain, is widely distributed, including Psb33s. Motif 19, which

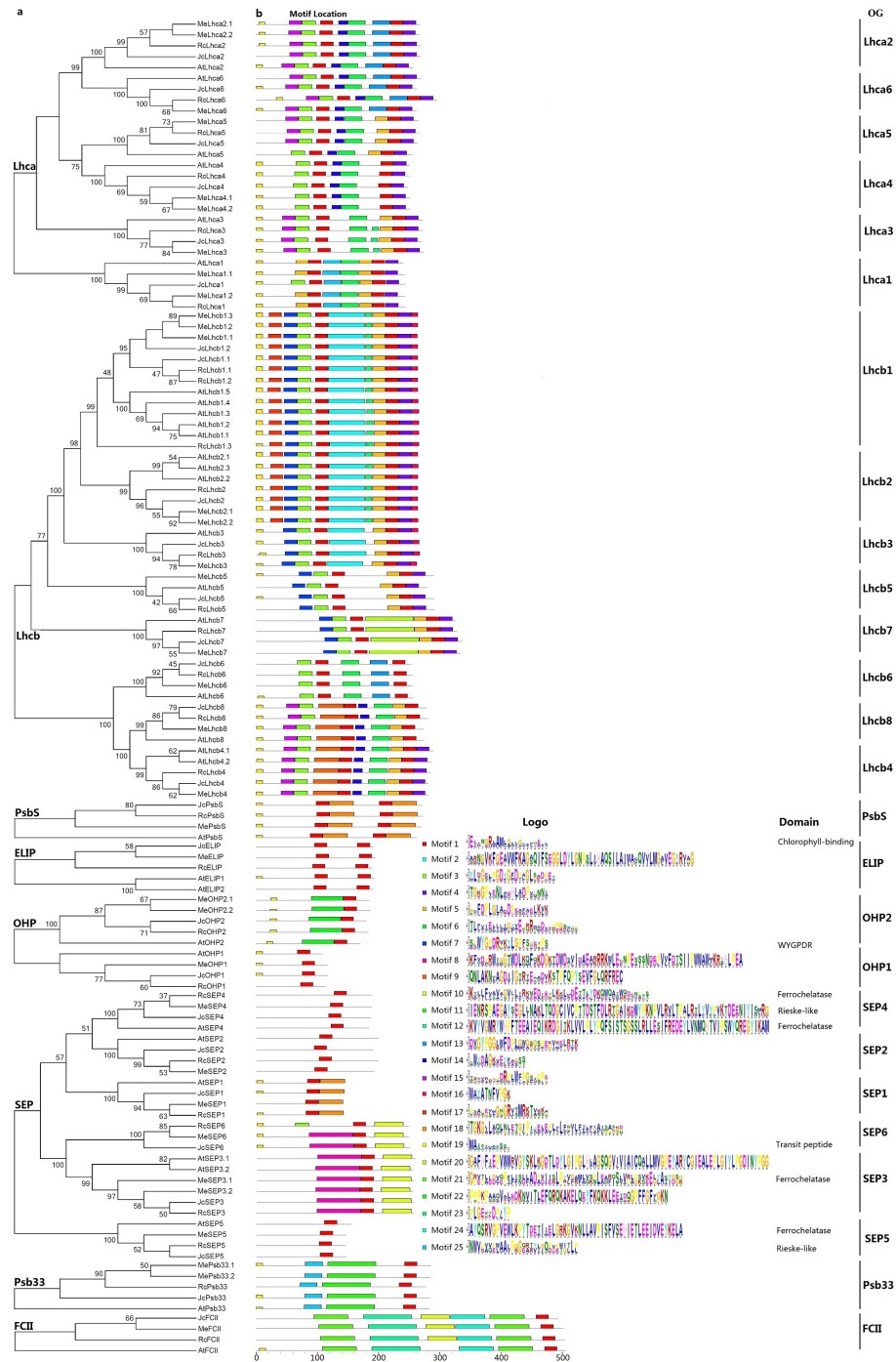

**Figure 3  Phylogenetic and conserved motif analyses of jatropha, castor, cassava, and arabidopsis Lhc superfamily proteins.** (A) Phylogenetic analysis of Lhca, Lhcb, PsbS, ELIP, OHP, SEP, Psb33, and FCII subfamilies; (B) Distribution of conserved motifs. Sequence alignment was performed using MUSCLE and unrooted phylogenetic trees were constructed using MEGA7 (maximum likelihood method; bootstrap, 1,000 replicates). Only bootstrap values at nodes supported by a posterior probability of ≥50% are given. The distance scale denotes the number of amino acid substitutions per site. The name of each OG is indicated next to the corresponding group. OG: orthologous group.

is characterized as chloroplast transit peptide, is also widely found. Motifs 25 and 11 are characterized as part of the Rieske-like domain (PF13806), where Motif 11 is Psb33-specific and Motif 25 is also present in Lhca1s. The Ferrochelatase domain (PF00762), which is FCII-specific, was shown to include Motifs 21, 12, 10, and 24. Among them, Motif 10 is also present in SEP3s and SEP6s. Motif 7, which includes the WYGPDR/WYGEER domain, is widely found in Lhcb1, −2, −3, −5, and −7 groups. WYGPDR has been proven to be essential for trimerization (*Hobe et al., 1995*; *Rogl & Kühlbrandt, 1999*), however, experimental evidence is still needed for WYGPD and WYGEER varieties. By contrast, little information is available for other motifs, including several group-specific motifs such as Motifs 8, 9, 18, 20, and 22.

Although these motifs are usually conserved within a certain OG, species-specific gain or loss was also observed. For example, the conserved Motifs 15, 23, and 10 are absent from AtLhca5, AtLhca3 or AtFCII, respectively, whereas the widely present Motif 8 in SEP3s and SEP6s is replaced by Motif 3 in MeSEP6 (Fig. 3).

## Expression profiles of *JcLhc* superfamily genes

Global gene expression profiles were investigated in various tissues, i.e., root (from 15-day-old seedlings), leafage (from 4-year-old plants, half expanded), leaf (mature leaf, fully expanded), IND (undifferentiated inflorescence of 0.5 cm diameter), PID1 (female flower with carpel primordia beginning to differentiate), PID2 (female flower with three distinct carpels formed), STD1 (male flower with stamen primordia beginning to differentiate), STD2 (male flower with ten complete stamens formed), and developing seed (19–28 days after pollination). Despite the expression of all identified genes, their transcript levels are highly variable over different tissues. As shown in Fig. 4, the majority of *JcLhc* superfamily genes are predominantly expressed in leaf, and the total transcript level of the whole superfamily in STD1, PID2, STD2, PID1, leafage, root, and seed accounts for 42.82%, 24.26%, 23.71%, 21.41%, 21.16%, 18.01%, 5.83%, or 1.75% of that in leaf, respectively. According to tissue-specific expression patterns, *JcLhc* superfamily genes can be divided into five main clusters: Cluster I is mostly expressed in leafage, including *JcPsbS*, *JcFCII*, and four *Lils* (i.e., *JcOHP2*, *JcSEP1*, *JcSEP2*, and *JcELIP*); Clusters II and III that contain the vast majority of *Lhc* family members and include approximately 63.00% of the whole superfamily are predominantly expressed in mature leaf, where Cluster III is also highly abundant in leafage (i.e., *JcLhca6*, *JcLhcb7*, *JcLhcb8*, *JcOHP1*, *JcSEP3*, *JcSEP4*, *JcSEP5*, and *JcPsb33*); Cluster IV that only includes *JcSEP6* is preferentially expressed in root, whereas Cluster V is typically expressed in STD1 (i.e., *JcLhcb1.2*, *JcLhcb2*, and *JcLhcb6*) (Fig. 4). It is worth noting that, *JcLhcb1.1* represents the most expressed gene in most tissues examined, implying its key roles.

Given drought and salt are two of the most important abiotic stresses affecting plant growth and development, photosynthesis, and crop yield, we thereby investigated the response patterns of *JcLhc* superfamily genes post drought or salt treatment in leaves and roots of eight-week-old seedlings. After withholding irrigation for 1, 4 or 7 d, the total superfamily transcripts in roots were not significantly changed, by contrast, initial increase followed by significant decrease were observed in leaves. The result is consistent with the

Zhao et al. (2020), *PeerJ*, DOI 10.7717/peerj.8465

**Table 2 Detailed information of 25 motifs identified in this study.** Motifs were identified using MEME.

| Motif | E-value | Sites | Width | Best match |
| --- | --- | --- | --- | --- |
| Motif 1 | 2.5e−1,988 | 205 | 21 | EJINGRLAMLGFLGFLVQEIL |
| Motif 2 | 6.2e−1,170 | 24 | 60 | ARNGVKFGEAVWFKAGAQIFSEGGLDYLGNPSLIHAQSILAIWACQVVLMGAVEGYRVAG |
| Motif 3 | 9.4e−992 | 69 | 23 | YLDGELPGDYGFDPAGLSADPET |
| Motif 4 | 8.6e−894 | 64 | 21 | TGKGPJENLADHLADPVHNNI |
| Motif 5 | 2.6e−582 | 58 | 21 | GSFDPLGLADDPEAFAELKVK |
| Motif 6 | 3.0e−546 | 40 | 29 | TLFVIELJLIGYVEFRRWADLDNPGSVYP |
| Motif 7 | 1.9e−374 | 32 | 21 | SPWYGPDRVKYLGPFSGETPS |
| Motif 8 | 4.2e−331 | 8 | 73 | KFVDPRWIGGTWDLKQFZKDGKTDWDAVIDAEAKRRKWLEENPESSSNDEPVVFDTSIIPWWAWIKRYHLPEA |
| Motif 9 | 1.4E−216 | 9 | 41 | QNLAKNVAGDIIGTRTEAADVKSTPFQPYSEVFGLQRFREC |
| Motif 10 | 3.5E−205 | 12 | 48 | KTLLFVAVAGVLLIRKNEDIETLKKLLDETTLYDKQWQATWKDZNPSS |
| Motif 11 | 9E−200 | 5 | 80 | IENRSPAEGAYSEGLJNAKLTQDGCIVCPTTDSTFDLRTGAIKDWYPKNPVLRVLTPALRTLYVYPVKTDEENIYISLRG |
| Motif 12 | 1E−170 | 4 | 80 | KVYVGMRYWHPFTEEAIEQIKRDGITKLVVLPLYPQFSISTSGSSLRLLESIFREDEYLVNMQHTVIPSWYQREGYIKAM |
| Motif 13 | 8.1E−166 | 13 | 29 | DVGYPGGLWFDPLGWGSGSPEKVKELRTK |
| Motif 14 | 3.6E−159 | 27 | 15 | SWYDAGKVEYFAGSS |
| Motif 15 | 1.8E−154 | 25 | 21 | TVCVKADPDRPLWFPGSTPPE |
| Motif 16 | 5.4E−153 | 24 | 11 | WAYATNFVPGK |
| Motif 17 | 7.6E−149 | 20 | 21 | PSAPEVMGNGRVTMRKTVKKA |
| Motif 18 | 8.6E−142 | 12 | 41 | TGKGLLAQLNJETGJPIYELEPLVLFNVLFALFAAINASKD |
| Motif 19 | 3.6E−140 | 80 | 11 | MATSTLAASSS |
| Motif 20 | 6.1E−140 | 4 | 80 | GAFHFIEPVWWRVGYSKLKGDTLDYLGIPGLHLAGSQGVIVIAICQALLMVGPEYARYCGIEALEPLGIYLPGDINYPGG |
| Motif 21 | 3.4E−142 | 8 | 57 | GPEPLLGVZPFLINLLADPVIERLPYVGAFLVKPLEAFISLVPLPKVEEGLASYGGG |
| Motif 22 | 2E−98 | 5 | 53 | PPPKPAAQVALDDKNVITLEFQRQKAKELQEYFKQKKLEETDQGPFFGFLGKN |
| Motif 23 | 4.3E−98 | 23 | 11 | PLGEVTDPJYP |
| Motif 24 | 4.6E−92 | 4 | 57 | AYQSRVGPVEWLKPYTDETIIELGRKGVKNLLAVPISFVSEHIETLEEIDVEYKELA |
| Motif 25 | 1.4E−106 | 10 | 29 | NWVPAVPLAALPGGZATYJGQPVPTGLLP |

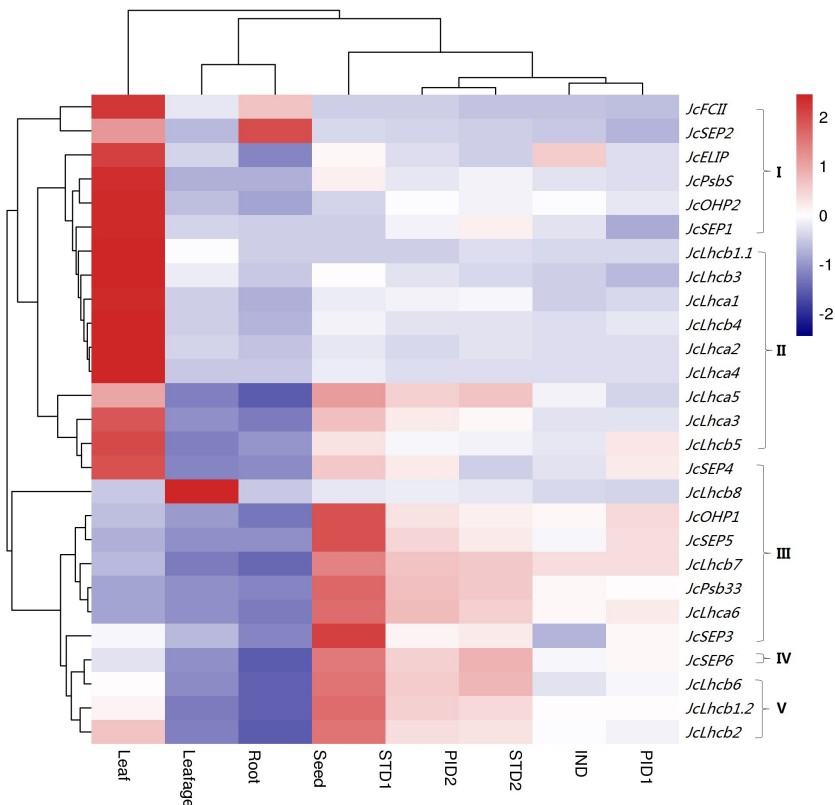

**Figure 4** **Tissue-specific expression profiles of *JcLhc* superfamily genes.** Color scale represents FPKM normalized $\log_{10}$ transformed counts where navy indicates low expression and firebrick3 indicates high expression. FPKM: fragments per kilobase of exon per million fragments mapped; IND, undifferentiated inflorescence of 0.5 cm diameter; PID1, female flower with carpel primordia beginning to differentiate; PID2, female flower with three distinct carpels formed; STD1, male flower with stamen primordia beginning to differentiate; STD2, male flower with ten complete stamens formed.

fact that the net photosynthesis rate (Pn) and stomatal conductance had decreased to 80% or 20% of those in the control after the start of the stress treatment for 2 and 7 d, respectively (*Zhang et al., 2015*). For 1 d, seven or three genes were significantly regulated in leaves and roots, respectively. Among them, *JcLhcb1.1*, *JcLhcb1.2*, *JcLhcb3*, *JcELIP*, *JcSEP2*, and *JcSEP5* were upregulated in leaves, whereas *JcOHP2* was downregulated; *JcLhcb3* and *JcFCII* were downregulated in roots, whereas *JcPsbS* was upregulated. For 4 d, ten or seven genes were significantly regulated in leaves and roots, respectively. Among them, *JcLhca4*, *JcLhcb2*, *JcLhcb4*, and *JcELIP* were upregulated in leaves, whereas *JcLhcb1.1*, *JcLhcb1.2*, *JcLhcb3*, *JcLhcb5*, *JcLhcb7*, and *JcLhcb8* were downregulated; *JcLhca3*, *JcLhca4*, *JcLhcb3*, *JcLhcb8*, and *JcPsbS* were upregulated in roots, whereas *JcLhca1* and *JcFCII* were downregulated. For 7 d, 19 or seven genes were significantly regulated in leaves and roots, respectively. Among them, *JcLhca4*, *JcLhcb2*, *JcPsbS*, *JcELIP*, *JcOHP1*, *JcSEP2*, and *JcSEP5* were upregulated in leaves, whereas *JcLhca3*, *JcLhca5*, *JcLhcb1.1*, *JcLhcb1.2*, *JcLhcb3*, *JcLhcb5*, *JcLhcb6*, *JcLhcb8*, *JcSEP1*, *JcSEP3*, *JcSEP4*, and *JcPsb33* were downregulated; *JcLhca1*, *JcLhcb1.1*, *JcLhcb1.2*,

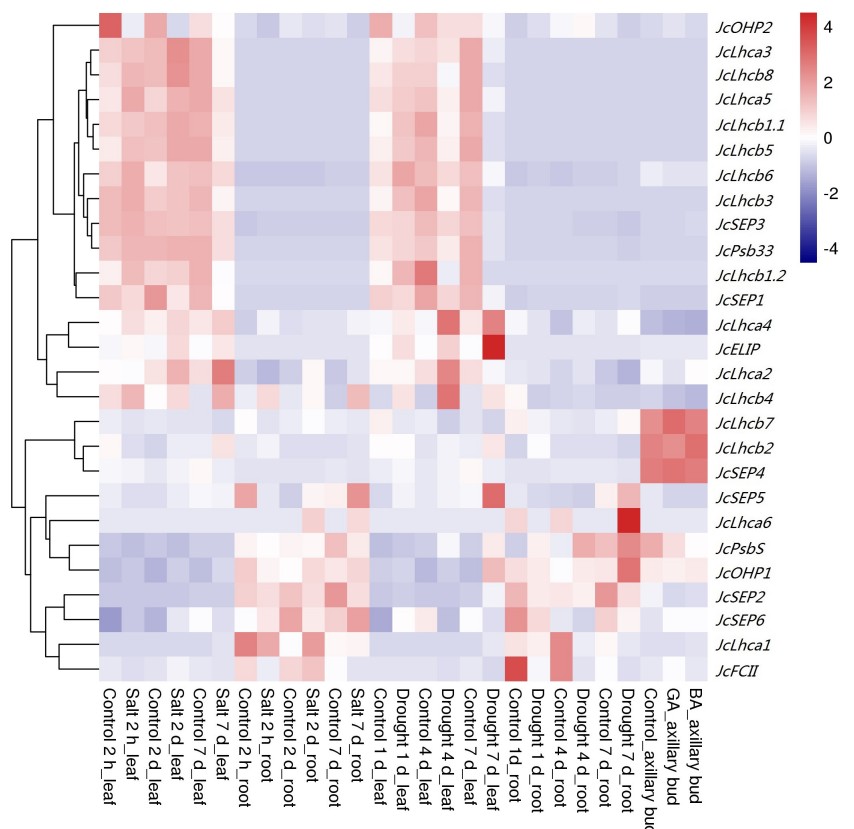

**Figure 5** **Expression profiles of *JcLhc* superfamily genes upon drought, salt, BA, or GA treatments.**
Color scale represents RPKM normalized $\log_{10}$ transformed counts where navy indicates low expression
and firebrick3 indicates high expression. RPKM, Reads per kilobase per million mapped reads.

*JcLhcb3*, *JcLhcb5*, and *JcFCII* were downregulated in roots, whereas *JcOHP1* was upregulated
(Fig. 5).

Similar to drought treatment, after applying 100 mM NaCl for 2 h, 2 d or 7 d, gradual
downregulation of total transcripts was only observed in leaves. For 2 h, five or three genes
were significantly regulated in leaves and roots, respectively. Among them, *JcLhcb2*, *JcOHP2*,
*JcSEP2*, and *JcFCII* were downregulated in leaves, whereas *JcLhcb1.2* was upregulated; in
roots, *JcLhcb8*, *JcSEP5*, and *JcFCII* were downregulated. For 2 d, six genes were significantly
regulated in both leaves and roots, respectively. Among them, *JcELIP*, *JcSEP2*, *JcSEP4*,
and *JcFCII* were upregulated in leaves, whereas *JcOHP2* and *JcSEP1* were downregulated;
*JcLhca1*, *JcLhca2*, *JcLhcb8*, and *JcSEP5* were upregulated in roots, whereas *JcLhcb1.1* and
*JcLhcb1.2* were downregulated. For 7 d, nine or seven genes were significantly regulated in
both leaves and roots, respectively. Among them, *JcLhca3*, *JcLhcb1.1*, *JcLhcb1.2*, *JcLhcb3*,
*JcLhcb5*, *JcLhcb8*, *JcSEP1*, and *JcSEP4* were downregulated in leaves, whereas *JcLhcb2* was
upregulated; *JcLhca3*, *JcLhcb1.2*, *JcLhcb3*, *JcLhcb5*, and *JcSEP5* were downregulated in roots,
whereas *JcELIP* and *JcFCII* were upregulated (Fig. 5). Downregulation of most regulated
genes is highly consistent with gradual decrease of Pn, where the Pn values of 2 and 7 d after stress treatment accounted for 83% or 50% of the control, respectively (*Zhang et al., 2014*).

Responses to gibberellin acid (GA) and 6-benzylaminopurine (BA) treatments were also examined in young axillary buds. Application of 10 μM GA for 12 h resulted in one upregulated (i.e., *JcFCII*) and two downregulated (i.e., *JcSEP2* and *JcSEP5*) genes, in contrast, no evident effect was observed for the same concentration of BA (Fig. 5).

## DISCUSSION

In green plants, the Lhc superfamily is consisted of four antennae protein families that play essential roles in light-harvesting and photoprotection (*Jansson, 1999*; *Klimmek et al., 2006*; *Zou, 2018*). Despite their importance, extensive research is still limited to the model plant arabidopsis and few other species such as *Chlamydomonas reinhardtii*, *Physcomitrella patens*, castor, and cassava (*Elrad & Grossman, 2004*; *Klimmek et al., 2006*; *Alboresi et al., 2008*; *Engelken, Brinkmann & Adamska, 2010*; *Zou, Huang & An, 2013*; *Zou & Yang, 2019a*). Among them, it's well established that cassava and arabidopsis experienced one or two additional WGDs after the so-called $\gamma$ hexaploidization event shared by all core eudicots: the recent WGD occurred in cassava is called $\rho$, which was estimated to occur within a window of 39–47 million years ago (Mya) (*Bredeson et al., 2016*; *Zou, Yang & Zhang, 2019*; *Zou & Yang, 2019a*; *Zou & Yang, 2019b*; *Zou & Yang, 2019c*), whereas two recent WGDs occurred in arabidopsis are known as $\beta$ and $\alpha$, which were estimated to occur within a window of 61–65 or 23–50 Mya, respectively (*Bowers et al., 2003*). From this point of view, analysis of species without recent WGDs may improve our knowledge on species-specific evolution of this special gene family. Jatropha, another economically Euphorbiaceous plant for potential biodiesel purpose, is a good candidate for such study. According to comparative genomics analysis, jatropha may share a common ancestor with cassava and castor at approximately 65 Mya (*Bredeson et al., 2016*), and no additional WGD occurred after their divergence.

In the present study, a first genome-wide identification and global analysis of *Lhc* superfamily genes were performed in jatropha, and the superfamily number of 27 members identified in this species is comparable to 28 reported in castor but relatively less than 35 or 34 present in cassava and arabidopsis, respectively (*Klimmek et al., 2006*; *Engelken, Brinkmann & Adamska, 2010*; *Zou, Huang & An, 2013*; *Zou & Yang, 2019a*). Nevertheless, 26 OGs representing four families (i.e., Lhc, Lil, PsbS, and FCII) were found in jatropha as observed in castor and cassava (*Zou, Huang & An, 2013*; *Zou & Yang, 2019a*). Few recent duplicates were identified in jatropha as well as castor, corresponding to no recent WGD occurred in these two species (*Chan et al., 2010*; *Wu et al., 2015*). Compared with castor that contains two dispersed duplicates, only one duplicate derived from tandem duplication was found in jatropha. By contrast, considerably more duplicates, i.e., nine, were identified in both cassava and arabidopsis (Table S1), reflecting the occurrence of one or two recent WGDs (*Bowers et al., 2003*; *Bredeson et al., 2016*). Despite having the same number of duplicates, cassava contains relatively more WGD duplicates (6 vs 5) but less local duplicates (3 vs 4), implying species-specific evolution pattern. Interestingly,

duplicates in both jatropha and castor are confined to the Lhcb1 group, however, in cassava and arabidopsis, local duplicates were also found in the Lhcb2 group (Table S1).

Among four families identified, both PsbS and FCII include a single member in four species examined in this study. By contrast, Lhc and Lil families are relatively complex. The Lhc family contains 14 OGs representing two subfamilies (i.e., Lhca and Lhcb), whereas the Lil family includes ten OGs representing four subfamilies (i.e., ELIP, OHP, SEP, and Psb33). According to crystal analyses, the Lhc family members usually contain three alpha-helixes, whereas the PsbS family features four (*Kühlbrandt, Wang & Fujiyoshi, 1994*; *Pan et al., 2011*; *Fan et al., 2015*). By contrast, one to three helixes were shown to be present in Lil proteins, i.e., one for OHPs and Psb33s, two for SEPs, and three for ELIPs (*Jansson, 1999*; *Engelken, Brinkmann & Adamska, 2010*; *Fristedt et al., 2015*; *Beck et al., 2017*). Similar results were also observed in this study, however, both JcSEP5 and JcFCII were shown to contain a single helix.

It is noteworthy that, among 26 OGs identified, SEP6 is absent from arabidopsis. SEP6 exhibits about 41.7%, 40.5% or 39.0–40.3% sequence identity with SEP3 in jatropha, castor and cassava respectively, implying their early divergence and species-specific gene loss. Indeed, SEP6 orthologs were broadly found in dicots, including *Carica papaya* and *Aquilegia coerulea* (*Zou & Yang, 2019a*). Additionally, Lhcb8 shows approximately 72.0%, 69.9%, 72.6% or 64.2–65.3% identity with Lhcb4 in jatropha, castor, cassava, and arabidopsis, respectively. Lhcb8 is widely present in core eudicots but not in *A. coerulea* and monocots, suggesting its recent origin. According to synteny analysis performed in arabidopsis, *Lhcb8* is more likely to be a duplicate of *Lhcb4* generated along the $\gamma$ event (*Bowers et al., 2003*; *Wang, Tan & Paterson, 2013*).

Potential roles of *JcLhc* superfamily genes could be inferred from their expression patterns and function-characterized orthologs in arabidopsis and other species. According to GO annotation, they belong to thylakoid membrane proteins that have activity of chlorophyll binding, pigment binding, xanthophyll binding, lipid binding, protein binding, iron-sulfur cluster binding, oxidoreductase, ferrochelatase as summarized in Table S4. Our transcriptional profiling not only supports the expression of all 27 *JcLhc* superfamily genes identified in this study, but also reveals key genes in a certain tissue, development stage or environment condition. Similar to that reported in arabidopsis (*Jansson, 1999*; *Klimmek et al., 2006*), genes encoding JcLhca1 to −4 and JcLhcb1 to −6, which are characterized as abundant Lhc proteins, were highly expressed in most examined jatropha tissues, especially in mature leaf. By contrast, four genes encoding so-called rare Lhc proteins (i.e., *JcLhca5*, *JcLhca6*, *JcLhcb7*, and *JcLhcb8*) are lowly expressed, exhibiting a similar expression pattern to members of Lil, PsbS, and FCII families. The result is consistent with our cluster analysis, which divides *JcLhc* superfamily genes into five clusters named I, II, III, IV, and V. These genes encoding abundant Lhc proteins belong to Clusters II and V, whereas rare *Lhc* genes were divided into Cluster III with the exception of *JcLhca5*. Clustering *JcLhca5* into Cluster II is due to its leaf-preferential expression pattern, but not the transcript level. Compared with mature leaf, leafage is considerably more sensitive to high light and other stresses. This is not surprising that the majority of members in Lil, PsbS, and FCII families are highly expressed in this special tissue, comprising Clusters I and III. It

is worth noting that, *JcSEP6*, the unique member in Clusters IV, is lowly expressed in most examined tissues with the exception of root. As a recently identified superfamily member, the detailed function of SEP6 still needs to be investigated. Furthermore, most *JcLhc* superfamily genes were regulated by drought and/or salt, two most important abiotic stresses affecting crop growth and yield (*Zhang et al., 2014*; *Zhang et al., 2015*). As expected, more genes are downregulated, especially for the leaf tissue, corresponding to decrease of transpiration rate and stomatal conductance (*Zhang et al., 2015*). Nevertheless, frequent upregulation of certain members was also observed, i.e., *JcLhca4*, *JcELIP*, *JcPsbS*, *JcSEP2*, *JcSEP5*, and *JcOHP1*. Interestingly, *JcLhca2* and *JcLhca5* exhibit distinct responses upon drought or salt stress, whereas *JcLhca2* was specifically regulated by salt and *JcLhca4*, *JcLhcb4*, *JcLhcb6*, *JcLhcb7*, *JcPsbS*, *JcOHP1*, *JcSEP2* and *JcPsb33* were only regulated by drought. The involvement of *Lhc* superfamily genes in stress response has been well documented in arabidopsis and other species, including high light, chloroplast retrograde signal, oxidative stress, abscisic acid, etc. (*Montané & Kloppstech, 2000*; *Staneloni, Rodriguez-Batiller & Casal, 2008*; *Gerotto et al., 2011*; *Tibiletti et al., 2016*). For example, ELIPs, as the name suggests, are induced by early and high light, as well as other stresses such as UV-B, cold, heat, drought, salt, hypoxia, and anoxia described in this study and elsewhere (*Heddad et al., 2006*; *Hayami et al., 2015*). In arabidopsis, knockout and overexpression of *ELIPs* resulted in decreased chlorophyll levels (*Casazza et al., 2005*; *Tzvetkova-Chevolleau et al., 2007*). Another high-light induced gene, *PsbS*, acts as the main sensor of the low pH in plants and plays an essential role in nonphotochemical quenching (*Bergantino et al., 2003*; *Li et al., 2004*; *Gerotto et al., 2011*; *Fan et al., 2015*; *Tibiletti et al., 2016*). OHPs, which are related to HLIPs in cyanobacteria, are essential for the formation of the PSII reaction center. In arabidopsis, mutations in *AtOHP1* or *AtOHP2* caused severe growth deficits, reduced pigmentation, and disturbed thylakoid architecture (*Beck et al., 2017*; *Hey & Grimm, 2018*; *Myouga et al., 2018*; *Li et al., 2019*). By contrast, two plant hormones (i.e., BA and GA), which can improve shoot branching after application to young axillary buds (*Ni et al., 2017*), had little effect on transcriptional regulation of *JcLhc* superfamily genes.

## CONCLUSION

This study presents a first genomics analysis of the *Lhc* supergene family in jatropha, resulting in 27 members that are distributed across nine out of 11 chromosomes. Despite a relatively smaller number of members, 26 orthologous groups representing four families were found, where SEP6 represents a novel group that has been lost in the model plant arabidopsis. Nearly one-to-one orthologous relationship was observed between jatropha and castor, however, species-specific gene expansion was observed in these two species as well as cassava and arabidopsis. Exon-intron structures, protein motifs, and expression profiles of *JcLhc* superfamily genes were also analyzed and discussed. These findings provide valuable information for further studies in jatropha and species beyond.

## ACKNOWLEDGEMENTS

The authors appreciate those contributors who make the related genome and transcriptome data accessible in public databases. The authors also thank the editor and two reviewers for their helpful suggestions.

### Funding

This work was supported by the National Natural Science Foundation of China (31700580 and 31971688), the Natural Science Foundation of Hainan province (319MS093), the Central Public-interest Scientific Institution Basal Research Fund for Chinese Academy of Tropical Agricultural Sciences (1630022019017 and 1630052017011), and the Research Fund of Guangdong University of Petrochemical Technology (2018rc55). The funders had no role in study design, data collection and analysis, decision to publish, or preparation of the manuscript.

### Grant Disclosures

The following grant information was disclosed by the authors:
National Natural Science Foundation of China: 31700580, 31971688.
Natural Science Foundation of Hainan province: 319MS093.
The Central Public-interest Scientific Institution Basal Research Fund for Chinese Academy of Tropical Agricultural Sciences: 1630022019017, 1630052017011.
The Research Fund of Guangdong University of Petrochemical Technology: 2018rc55.

### Competing Interests

The authors declare there are no competing interests.

### Author Contributions

- Yongguo Zhao performed the experiments, analyzed the data, and approved the final draft.
- Hua Kong and Yunling Guo performed the experiments, authored or reviewed drafts of the paper, and approved the final draft.
- Zhi Zou conceived and designed the experiments, performed the experiments, analyzed the data, prepared figures and/or tables, authored or reviewed drafts of the paper, and approved the final draft.

### Data Availability

The data is available in the Supplementary Files.

### Supplemental Information

Supplemental information for this article can be found online at http://dx.doi.org/10.7717/peerj.8465#supplemental-information.

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
