# Peer review of "Light-harvesting chlorophyll a/b-binding protein-coding genes in jatropha and the comparison with castor, cassava and arabidopsis"

_PeerJ, doi:10.7717/peerj.8465_

## Round 0.1 · original submission · Major Revisions

Please address all issues highlighted by the reviewers and remove the excessive number of self-citations, which should instead, as reviewer#2 pointed out, refer to the original methods papers.

Reviewer 1 ·

Basic reporting

no comment

Experimental design

no comment

Validity of the findings

no comment

Additional comments

The results presented are mainly descriptive, and the results are lack of further evidence to be supported.

Although sufficient data were provided, the major problem is the following.
(1) The results are relatively simple, and the main figures didn’t provide enough information. The authors identified genes by BLAST searching. Then the authors performed hierarchical cluster based on FPKM values. I feel the work didn’t use very strong evidence to tell what the expression pattern and molecular biology function of the genes. I suggest the authors test expression levels of genes in figure 4, and 5 using qRT-PCR.
(2) The authors performed a synteny analysis in the text. However, no images in the main text related were shown.

Some minor concerns are as the following.
(1) Log2 based values are observed in line 121, while log10 scale are observed in other parts.
(2) Line 183. Arabidopsis should be italic.
(3) Line 250. Was should be were.
(4) Line 256. light harvesting should be light-harvesting.
(5) Line 280. Subfamiles should be subfamilies.

Reviewer 2 ·

Basic reporting

No comment.

Experimental design

The submitted article is within the scope of PeerJ. The study was well-conducted, and the results, especially for the gene identification and annotation, are quite solid.
However, many methods used are not fully described, in favor of pointing out to other articles. This is the case for instance on lines 96, 105, 107, 113, 116-117, 118, 122. The authors should at least reduce the number of these references and give a more thorough description of their methods.

Validity of the findings

No comment.

Additional comments

I only have a few other minor comments:
1. Information about the functions associated to the conserved motifs can be found in the lines 191-200; this important information should also be added to Figure 3, where the motifs are depicted without further information.

2. Concerning the expression profiles of the genes: in lines 215-219, the authors state that the genes can be divided into 5 clusters, depending on their tissue-specific expression pattern. A reminder of the genes, or at least gene families, present in each cluster would be welcome.
Also, do these clusters correspond to functional division? For instance, Cluster I, with a strong expression in leafage, seems to contain mainly stress genes, while the main light harvesting antennas of PSI and PSII are divided between Clusters II and V.
Developments of this part, in the results and/or discussion, would thus be welcomed.

3. In the discussion, the lines 305 to 319 are just an overview of the role of some genes in stress response in general. A few lines on the role these genes might play in the particular stresses studied here (salt and drought) would be welcome.

---

## Round 0.2 · Minor Revisions

Please address the few remaining concerns.

Reviewer 2 ·

Basic reporting

No comment.

Experimental design

No comment.

Validity of the findings

No comment.

Additional comments

The manuscript is improved.
However I still have two minor concerns:

1. I appreciate the effort of the authors into supplying more info on the motifs in figure 3. However I believe that the functional information provided in the text in lines 210-220 in the track-changed manuscript (attribution of the motifs to CB domain, transit peptide, Rieske-like domain…), even though not available for all motifs, would be better suited and more understandable to non-specialist readers than just representative sequence in that figure.

2. Previously my last comment was the following: “In the discussion, the lines 305 to 319 [now lines 352-368 in the track-changed manuscript] are just an overview of the role of some genes in stress response in general. A few lines on the role these genes might play in the particular stresses studied here (salt and drought) would be welcome.”
Although the authors accept this comment and write that the manuscript has been revised, I cannot find this revision in the text: the section is identical to the previous version, and the only change to the discussion concerns the cluster analysis and makes no mention of salt and drought stresses.

I am satisfied with the other answers and revisions.

---

## Round 0.3 · Minor Revisions

There is still some experimental detail missing: no detail is given for the statistical part of the RNAseq analysis. What program or procedure were used to find the differentially expressed genes? Giving the FDR and logFC thresholds is not sufficient

Reviewer 2 ·

Basic reporting

No comment.

Experimental design

No comment.

Validity of the findings

No comment.

Additional comments

I am satisfied with the modifications of the manuscript, and have no further comments.

---

## Round 0.4 · Major Revisions

Thank you for addressing my latest request. Upon conferring with the Section Editor responsible for the Plant Sciences section in PeerJ, we reached the conclusion that in spite of your efforts there are still several unacknowledged limitations and areas for improvement in your manuscript.Specifically, the super-family described in this paper is well studied in model plants, and in other representative systems. It is not explained, or reasoned why this study would address contribution to the model works or why this select species is a good candidate for this study; the manuscript still looks too much like a standard bioinformatics script run analysis of a selected gene set with no consequence of study developed. There was no apparent match between the reason the plant was selected (namely biofuels) and the gene set that was analyzed (light capture and stress). There were no experiments developed to test intense enough for any of these attributes, or at least the intentions were poorly developed and weak in approach. The description of the gene expression analysis was poorly described and basically without motive.

As studies included tissue and treatment breakdowns, annotations would be needed to build this work in with well studied model systems, being able to link genes to counterparts within the models. Journal manuscripts are often scanned by text-mining software that locates and extracts core data elements, like gene function. Adding standard ontology terms, such as the Gene Ontology (GO, geneontology.org) or others from the OBO foundry (obofoundry.org) can enhance the recognition of your contribution and description. This will also make human curation of literature easier and more accurate

---

## Round 0.5 · accepted · Accept

Although the authors have not included all of the requested background information regarding the choice of plant , I do think that the manuscript is a sufficient addition to the literature.